# Seeking Health Information: A Qualitative Study of the Experiences of Women of Refugee Background from Myanmar in Perth, Western Australia

**DOI:** 10.3390/ijerph19063289

**Published:** 2022-03-10

**Authors:** Georgia Griffin, S. Zaung Nau, Mohammed Ali, Elisha Riggs, Jaya A. R. Dantas

**Affiliations:** 1Curtin School of Population Health, Curtin University, Bentley, WA 6102, Australia; m.ali@curtin.edu.au (M.A.); jaya.dantas@curtin.edu.au (J.A.R.D.); 2School of Management and Marketing, Curtin University, Bentley, WA 6102, Australia; z.nau@curtin.edu.au; 3Murdoch Children’s Research Institute, Parkville, VIC 3052, Australia; elisha.riggs@mcri.edu.au; 4Department of General Practice, University of Melbourne, Parkville, VIC 3000, Australia

**Keywords:** apps, communication, community, health information, health literacy, health promotion, Myanmar, refugee, trauma, women

## Abstract

Women of refugee background are subject to significant health inequity. Access to health information and a good level of health literacy are integral components to manage one’s health needs. The aim of this study isto understand the experiences of women of refugee background from Myanmar seeking and accessing health information. Semi-structured interviews were conducted with 14 women of refugee background from Myanmar resettled in Western Australia. Interpretative phenomenological analysis underpinned the study and was conducted on the interview data. Three superordinate themes and nine subordinate themes emerged from the analysis: (1) Seeking health information (Motivation and Sources), (2) Facilitators and Barriers (Communication, Navigating the system and Community) and (3) Seeking health information in the context of past experiences (Health information as a by-product of healthcare, Health professionals’ provision of health information, Accessibility of healthcare and Expectations on resettlement). These themes provide insight into the challenges of accessing understandable and actionable health information and of promoting the health literacy of women of refugee background from Myanmar. Co-designed community-based and health service interventions should be trialled, including trauma-informed training for health professionals, health information apps and community health promotion programs. Community engagement, participation and evaluation are critical for determining the effective interventions to address the inequalities experienced by this population.

## 1. Introduction

Globally, the number of forcibly displaced people is at its highest documented level. Conflict, threats to human rights, the COVID-19 pandemic and climate change events have resulted in over 1% of the global population being forced to flee their homes [1]. Recently, in Myanmar, conflict and instability associated with the February 2021 military coup, ongoing food insecurity and successive waves of the COVID-19 pandemic have resulted in school closures, reduced healthcare access and increased gender-based violence [1,2]. Poverty is expected to further increase in 2022, therefore increased displacement of people from Myanmar is likely [1].

Decades of political instability, violence and persecution in Myanmar have resulted in over one million people being displaced or seeking asylum in nearby countries, including Thailand, Malaysia and Bangladesh [1,2,3,4]. Myanmar is composed of numerous ethnic groups, including the Chin, the Karen, the Kachin, the Mon and the Rohingya [3]. Currently, over 900,000 Rohingya refugees from Myanmar live in congested refugee camps in Cox’s Bazaar, Bangladesh [1]. For people living in these refugee camps, the COVID-19 pandemic has exacerbated pre-existing socioeconomic problems, including access to health services, gender-based violence and employment opportunities. Climate events, including cyclones, floods and fires, have also affected people living in the refugee camps.

After years of precarious living in refugee camps or unofficial resettlements, people identified as refugees may resettle in third countries through the United Nations High Commissioner for Refugees (UNHCR) Resettlement Program [3,5]. The most common resettlement countries for people from Myanmar are the United States of America, Australia, Canada, New Zealand and South Korea [5]. The number of people who resettle in Australia through the UNHCR Resettlement Programme is determined by the Australian government each year. In 2021–2022, 13,750 places wereavailable for people to resettle in Australia through this program [6].

On resettlement, people of refugee background may have complex resettlement and health needs [7]. Women of refugee background from Myanmar often have complex health needs because of prior violence, malnutrition, infectious diseases, perinatal complications and mental health trauma [8,9,10,11]. Women of refugee background typically experience poor health service access and health outcomes on resettlement [7,12]. Inadequate health literacy and health knowledge have been identified as factors in poor health service access for people of refugee background, including that of women from Myanmar [12,13,14]. Inadequate health literacy has been associated with difficulties in communicating and understanding information about health, making appointments and locating health services [13,14,15,16,17].

Health literacy is a key factor in health outcomes [18]. While many definitions of health literacy exist, it is essentially “the literacy and numeracy skills that enable individuals to obtain, understand, appraise, and use information to make decisions and take actions that will have an impact on health status” [18] (p. 161). These skills have the potential to empower individuals to manage their own health needs [19]. Individuals draw upon health literacy skills to access and use information about their health or health services in a meaningful way, facilitating their access to health services and management of their health needs [18]. In their study with a Bhutanese community of refugee origin, Chao and Kang [20] found that health literacy emerged as a socio-cultural concept. English language proficiency, community health experiences and engagement with community were found to be mediating factors of health literacy. People of refugee background from Bhutan learnt from one another’s experiences of health and healthcare. This suggests that measures of and interventions to improve health literacy should consider the socio-cultural context of the community.

While previous research has explored how women of refugee background from Myanmar access healthcare and their healthcare experiences [13,14,15,16,17], little is known about the women’s experiences of seeking and obtaining health information, that is, information about health, illness or health services. This study seeks to address this gap in the literature by offering insight into the experiences of women of refugee background from Myanmar seeking and accessing health information. These experiences can offer crucial insight into the women’s application of health literacy skills and how they may be supported to obtain health information and develop these skills.

## 2. Materials and Methods

### 2.1. Aim

The aim of this qualitative study was to explore the experiences of women of refugee background from Myanmar seeking and accessing health information. The following objectives guided data collection and analysis:To identify and explore how women of refugee women from Myanmar access health-related information;To identify facilitators and barriers to accessing health-related information;To compare the pre- and post-migration experiences of women of refugee background accessing health-related information.

### 2.2. Context of the Study

This study was conducted in Perth, Western Australia. According to the 2016 census, Myanmar was the most common country of origin of people who had resettled in Western Australia on a humanitarian visa [21]. On resettlement, people who have entered Australia on a humanitarian visa are referred to the Humanitarian Entrant Health Service for cost-free holistic health screening [22]. During this screening, referrals to services based on the identified health needs are made. People may be referred to the Humanitarian Settlement Program, which supports them to navigate health services amongst other resettlement needs [23]. Health service eligibility may vary dependent on visa type [24]. Despite these support systems, difficulty accessing and navigating health services persists [12].

On 1 February 2021, the military junta seized power in Myanmar; violence quickly followed [25]. For the Myanmar diaspora in Western Australia, the coup has caused significant distress. To support loved ones in Myanmar and condemn the actions of the military, fundraising events were quickly organised. On the advice of the Burmese bicultural researcher, study participants were recruited at such events, along with Church services and community education sessions.

### 2.3. Community Engagement

Community engagement was an essential component of the study design and for data collection. An active member of the Myanmar community in Western Australia, the co-author, S.Z.N., promoted the research and enabled access to the community. As a cultural insider, a bicultural researcher is able to offer insight into cultural nuances [26]. She and the first author (G.G.) facilitated community engagement by visiting community and Church-based groups and events at the invitation of community leaders. When attending community events, community leaders often also assigned a guide to accompany the first author and introduce them to eligible women, to promote the women’s comfort.

### 2.4. Interpretative Phenomenological Analysis

In order to gain insight into the experiences of women of refugee background from Myanmar, interpretative phenomenological analysis (IPA) was selected. This approach was chosen to enable the research team to gain insight into how women of refugee background from Myanmar make sense of the phenomenon of seeking health information [27] in the context of factors that may influence their experiences, such as gender or refugee identity [28,29]. The focus of IPA on the lived experience of the individual makes it well suited for use in studies with people of refugee background [30]. Furthermore, as each participant tries to make sense of their experience seeking health information, the researcher tries to understand the meaning that the participant attributed to the experience [31].

IPA guided the study design, including interviews as the data collection tool, the interview guide and analysis of the women’s experiences. Interviews are a well-established data collection tool in IPA, allowing for the collection of rich data [32]. Sample sizes are typically small in IPA to allow the researcher to focus on the experiences of the individuals [27,31]. Fourteen women participated in this study. The interview guide was designed to enable the exploration of the women’s experiences and the meaning that they attributed to these experiences. The participants were encouraged to lead the direction of the interview [31].

### 2.5. Data Collection

Purposive and snowball sampling approaches were utilised, established sampling methods in cross-cultural research [33]. This study was advertised by the bicultural researcher, community leaders and the participants themselves. Inclusion criteria included female gender, originally from Myanmar, of refugee background, being at least 18 years of age and able to provide informed consent. Women who were interested in participating in an interview provided their contact details to S.Z.N. and G.G. Women were then phoned or emailed to arrange an interview. A total of twenty-three women were contacted; fourteen withdrew or were not contactable. Additionally, five women volunteered to participate through the bicultural researcher. In total, fourteen women participated in an interview. Small sample sizes are recommended for interpretative phenomenological analysis due to the idiographic nature of this approach [27,31].

### 2.6. Interviews

Data werecollected through semi-structured interviews from June to August 2021. Women chose the language they would like the interviews conducted in. The bicultural researcher co-facilitated seven interviews in Burmese, and Burmese and English, assisting with interpretation and understanding of cultural nuances. Four women were interviewed with the assistance of a Hakha Chin interpreter present as they spoke a different language to the bicultural researcher. The same interpreter assisted with all four of these interviews for continuity. She had previous experience interpreting in healthcare settings, was briefed on the study prior to the interviews and encouraged to share her cultural insights to assist the interview process and to establish her positionality [34]. Three women with high English language proficiency chose to share their experiences in English without the bicultural researcher present. Twelve of the interviews were conducted in the women’s homes and two of the interviews were conducted over a videocall; the women chose these locations as the easiest and most comfortable setting for them.

An interview guide comprising fourteen open-ended questions was used. Women were asked to share experiences of seeking health information on resettlement in Australia and prior to their resettlement, in refugee camps or Myanmar. They were asked to reflect on what made it easy or difficult to find health information in these settings, and if there were times when they could not seek health information when they wanted it. Questions included, “*Can you please tell me about an experience of finding out information about health or a health service in Australia?”* and “*Where did you look for health information before arriving in Australia?”*. The order of questions was flexible to allow the exploration of the women’s experiences. Prompts, such as “*can you give me an example”,* were used to encourage women to provide more detail.

### 2.7. Data Analysis

All bilingual interviews (Burmese and Hakha Chin) were interpreted and translated into English during the interview and audio recorded. The English language data weretranscribed verbatim by the first author and entered into NVivo software [35] for data management and to assist with analysis. Hand-written notes from the two interviews that were not recorded were also included in the dataset.

Interpretative phenomenological analysis of the interview data was conducted by the first author (G.G.) according to the following steps: (1) Reading and re-reading, (2) Initial noting, (3) Developing emergent themes, (4) Searching for connections across emergent themes, (5) Moving to the next case, (6) Looking for patterns across cases, and (7) Taking interpretations to deeper levels [32]. Each interview was analysed, revealing emergent themes, before moving onto the next interview. Once this process was complete, patterns across the cases were identified. These were taken to deeper levels, ultimately presented as superordinate and subordinate themes containing significant quotes from the women and a detailed interpretative summary [31].

### 2.8. Reflexivity and Rigour

The interviews were conducted by the first author; seven interviews were co-facilitated by the second author and bicultural researcher and four interviews were conducted with the assistance of an interpreter. To maintain objectivity, the first author and bicultural researcher adopted a reflective stance, debriefing following the interviews conducted together. The involvement of a researcher of a shared culture and language with the participants enhanced rigour in the cross-cultural research [26]. The first author made reflexive notes following the other interviews.

### 2.9. Ethical Considerations

Ethical approval was granted by the Curtin University Human Research Ethics Committee (HRE2020-0721). Each participant signed a consent form in English or Burmese to proceed with the interview. For those that were not literate in either of these languages, the consent form was read to them. The women were advised that their participation was voluntary, that they could withdraw from the interview at any point and that their relationship with their community group would not be affected by their decision to participate or not participate in the study. Pseudonyms were allocated to participants and used when reporting the results to ensure their anonymity.

Research involving people of refugee background involves specific ethical considerations [30]. Interviews with people of refugee background can potentially re-traumatise them as past traumatic experiences may be discussed [36]. The first author sought to avoid discussing potentially re-traumatising topics, such as violence. However, women shared prior health experiences, including those in which they felt their lives were threatened or distressed by not having their health needs met. One woman became teary. She was asked if she would like to stop or pause the interview. However, she chose to continue as sharing her experiences was important to her. It is important to recognise and support the each individual’s right to self-determination, that is, their choice to participate in this research [36]. The researchers offered mental health support; however, she felt she did not require additional support as she was already engaged with a mental health service.

## 3. Results

Participant demographics are presented in Table 1. In total, 14 women participated in semi-structured interviews. Two participants, sisters, requested an interview together. One sister later participated in an individual interview at her request because she felt she had more to share. Seven women of Chin ethnicity, three women of Kachin ethnicity, two women of Mon ethnicity and two women of Karen ethnicity participated. Two of the Chin women elected not to have their interviews audio-recorded; handwritten notes were taken instead. Interview length ranged from approximately 10 min to 1 h with an average of 35 min.

Women’s ages ranged from 25 to 61 years. They had been living in Australia from 1 year and 3 months to 15 years, with a mean period of resettlement of 8 years. All women cared for children, grandchildren or elderly parents. Three of the women had completed primary school, three had completed high school, five had completed a Technical and Further Education (TAFE) qualification, while only one had completed university. The remainder of the women elected not disclose their level of education. Three of the women were engaged in paid employment, two did not disclose their employment status, two were studying and the remainder were not employed. Two of the Chin women were sisters, and two other Chin women were sisters-in-law.

Three superordinate themes and nine subordinate themes related to the objectives of this study emerged during analysis. These are presented in Table 2.

In the first superordinate theme, Seeking health information, the women’s motivation to seek health information and the sources from which they seek it are explored. The second superordinate theme, Facilitators and barriers, centres on barriers to women accessing, understanding and communicating with health professionals, reflecting health professionals as their primary and preferred source of health information. Women shared how barriers can be overcome and the role of community in facilitating access to health information. In the third superordinate theme, Seeking health information in the context of past experiences, women were asked to reflect on seeking health information in the pre- and post-migration context. They shared experiences of seeking or not seeking healthcare, and the challenges they faced to manage their own health. These experiences formed the context from which they sought health information once resettled in Australia.

The superordinate and subordinate themes are described in the following text. Supporting quotes are presented in italics. If the quote was interpreted by the bicultural researcher or interpreter, a postfix was added to show this (‘B’ for bicultural researcher and ‘I’ for interpreter).

### 3.1. Superordinate Theme: Seeking Health Information

#### 3.1.1. Subordinate Theme: Motivation

Women were motivated to seek health information by need. Such needs could include pregnancy, feeling unwell including chronic and complex health needs, and the needs of children or other family members. Aye Aye explained:
… *she only seek the information when she’s not well, only for that particular one. She’s not researching… because she feel that if she search the information beforehand, she may start feeling like… she is not well… she only seek or go to the GP once she is not well… and search the information after*. (Aye Aye^B^)

Pregnancy was identified as a reason they had sought health information by eight women. Of these, five recounted stories of no antenatal care and complications of pregnancy, including pregnancy loss and premature deliveries. Chaw shared the story of her pregnancy with her daughter who was born two months premature.

*…I didn’t know that I have to go to the hospital and then you know, get a regular check… at about seven month of pregnancy, um I had a, you know, tummy pain, so I went to, to the hospital*. (ChawI)

She went on to explain how this experience motivated her to seek out health information.

*So I was blaming myself, it’s because of me. So after that… I started looking for, you know, informations [sic], try to get some knowledge about health so*. (ChawI)

Some women expressed a desire to learn about how they could maintain or improve their health. Specifically, women wanted more information on sexual and reproductive health, healthy eating and how to access help in an emergency. Two women described seeking out such information themselves.


*I like to know about my health as well. If you know something is uh, you um, sometimes you don’t know what you, what happen in your body… So I like to make sure. We can protect it.*
(NandarB)

#### 3.1.2. Subordinate Theme: Sources

Health professionals were women’s primary source of health information and often their preferred source. The general practitioner (GP) was a frequently cited source of health information. Other health professionals cited included the community health nurse, school health nurse, midwife and hospital-based doctors and nurses.


*…So I prefer to get information from the doctor…*
(NilarI)

While for some women this was a preference, for others health professionals were the only source of health information known or deemed accessible.


*So she doesn’t know how to search… So the only way she could [access health information] is the GP.*
(TinB)

Health professionals, such as nurses, could be approached opportunistically when women had health information needs that had not been met elsewhere. For example, when Aye Aye was unsuccessful obtaining information about her daughter’s hip problems from the specialist, she asked a community nurse for information during an appointment for her other child.


*So actually she went there for that little boy. And then she just had the conversation… Otherwise she would not have known about it…*
(Aye AyeB)

Nine women sought health information from the Internet. They primarily used the Internet to seek information about how to access a health professional to engage with in person, such as through HealthEngine, an online search engine designed to find and make appointments with health professionals. Some women also accessed health information videos published by doctors in Myanmar over social media or Burmese language websites. Women described varying degrees of confidence in their ability to find what they needed on the Internet. Some wanted reliable websites where they could trust the information.


*… I read a lot of things… I would prefer if I am know trustworthy source or a link or a website that I can go and look for certain topic.*
(SuSu)

Women discussed their health with family and friends, including friends outside of the Myanmar community. Sharing experiences with friends could motivate women to seek further health information.


*…sometime when we have conversation, then we say like, oh, you know, I didn’t know I had this until I go and check with my GP and I’m like oh okay, that’s interesting… So I can hear their stories and experience are… prompt me to… do some research of my own.*
(Win)

### 3.2. Superordinate Theme: Facilitators and Barriers

#### 3.2.1. Subordinate Theme: Communication

Communication emerged as a significant barrier to obtain and understand health information for every woman. This reflected women’s preference to obtain health information by conversing with people, primarily health professionals. The communication barriers identified were numerous and complex, including language, women not asking questions and perceived negativity from health professionals.

Women were alert to signs or cues that their questions may not be welcome. Feeling rushed and negative body language contributed to women not feeling comfortable to ask questions. For example, Aye Aye described an experience in which
*…the doctor was so angry and… teared [sic] the paper… So she feel like um, you know, there was a misunderstanding because of the language barrier… and she, you know, feel a bit scared to ask more question there… That’s why she didn’t ask anymore.*(Aye AyeB)

Other times, finding the right word or communicating about specific problems were described as too challenging and women did not ask questions. For women who felt confident in their English language skills, medical terminology or words for body parts were not always known, thus posing a barrier to communicating about health needs. SuSu described the challenges of describing symptoms for health professionals for which she did not have an English equivalent.


*Sometimes you don’t have the word in English, that the Karen have… And you don’t have the words in Karen that English have so that makes it very difficult with some disease, some… problem.*
(SuSu)

When communication broke down, women were alert, watching body language and observing others to piece together the information they required. Sometimes, however, women did not overcome these barriers.

Women sometimes described the presence of an interpreter as equal to understanding health information. However, women reported experiences with interpreters in which they were not satisfied with the communication, perceiving information to be omitted or mistranslated. Other times, they were challenged to communicate with an interpreter who did not speak their primary language. Phyu explained:
*…whenever she go and seek for the uh, medical advice or informations, uh it will be better that the doctor understand that they need the translations… without any assumption that all the people from Burma will understand Burmese.*(PhyuB)

Interpreters were not always engaged when needed, however, rather engaging if language became a significant barrier when trying to communicate or if the health need was deemed to be serious enough by the woman. One woman described her approach:


*…when things get hard… she get the interpreter…when like a minor health issues, she just talk.*
(ThidaB)

To overcome the language barrier, women sought out health professionals with Burmese names or who may speak Burmese. However, if the woman perceived that she was not receiving the information or treatment that she required, she found a different GP, Burmese or non-Burmese speaking. Women also brought family members, including their children, or Burmese-speaking caseworkers with them to act as informal interpreters. Cultural stigma could hinder communication when seeking sexual and reproductive health information. As Win explained, she did not discuss her sexual and reproductive health information needs with her GP due to *culture barrier and confidential things* (Win); she did not realise her discussions with her GP would be confidential from her Karen community.

Written information was described as helpful, in part because it could be checked later. Words that were not understood could be checked with a dictionary or explained by friends or family members. However, women wanted information in their primary language, Chin for example. Demonstrations, pictures, videos, short simple sentences, verbal explanations of the written information provided and telling a story to explain a piece of information were all identified as factors thatfacilitated their understanding of health information.

#### 3.2.2. Subordinate Theme: Navigating the System

Women described challenges navigating the health system. As health professionals were their primary source of health information, seeking health information could require them to identify an appropriate health professional, make an appointment, locate the health service, arrange transport and attend the appointment. Each of these steps could be presented as a barrier to obtaining health information. Home visits by health professionals, such as midwives or community nurses, were identified as particularly helpful, as they negated some of these barriers for women.


*So like the visit from the community nurse for her parents will be better… she has to take care of both of them and it’s a bit of struggle… So instead of going and you know, seeking to the family doctor all the time, if someone can come and check, that would be great.*
(TinB)

Caseworkers, support workers and family members were key figures on arrival for explaining how the health system worked, how to make appointments and identifying a specific GP or the multicultural women’s health service. If women could not recall these services being mentioned by the caseworker, they could be lost. Some women had complex and chronic health needs and could be responsible for caring for children or relatives. Tin described the difficulties she faced when seeking health information to manage her own complex chronic health needs and those of her child:
*So there are three things. First is the language. Uh so that they need support for the translation. And the second one if transport and mobility, later adding, …another difficulty add on is the affordability because it cost a lot for them.*(TinB)

Women learnt from experiences navigating the Australian health system. Phyu, for example, described having financial support withheld because she fell behind on her child’s immunisations. As a humanitarian entrant, Phyu was eligible for financial support from the government; however, a condition of this supportwas that her child have her immunisations as required by the National Immunisation Program Schedule. Phyu was unaware of this schedule and consequently did not receive a financial support payment. To prevent this happening again, she sought information from her GP who introduced her to the government website, myGov, which enabled her to access her ownand her child’s health records and stay uptodate with the immunisation schedule. She spoke with pride about her use of this website to access health information.


*…She miss the um, the immunisation schedule for her child because she wasn’t aware of it. Then [Government Agency] pen-, penalise for it… she could not even figure it out, why… So she ask her GP… when um, the child needs to get immunisation and then the GP recommend her to open the myGov account.*
(PhyuB)

Difficulty navigating the system and a lack of knowledge about available health services left women vulnerable when their health needs were not met. For example, Thida described visiting her GP repeatedly due to antenatal bleeding. Although she was not satisfied with the GP’s reassurances that this was normal, she did not know how else to obtain health information about bleeding during pregnancy or alternate health services. Eventually, severely unwell, a friend took her to the emergency department of a tertiary women’s hospital where she delivered her daughter prematurely. From this experience, she developed a formula for managing her health needs.


*So she has a fixed formula now. Uh, it’s the general illness or the sickness, she will go either GP or… medical centre… Otherwise if it’s a woman-related health issue, she will go to uh [multicultural women’s health service]. And something urgent, the emergency.*
(ThidaB)

#### 3.2.3. Subordinate Theme: Community

Community networks could assist to overcome barriers to accessing and understanding health information. One woman explained,
*Okay so, oh I don’t know so, I just ask around all my friends, how I can get some, you know, get health or, or get some information about health.*(NilarI)

Women typically referred to a key figure or figures, often someone who had migrated before they had, such as a sister-in-law, brother or husband. Key figures assisted with overcoming practical barriers to accessing health professionals or health services. Sometimes, this was as simple and pivotal as explaining what a GP is and how to access one. Sometimes, this was about providing transport, interpreting or simply accompanying them. If family members or community were not able to assist with transport for example, barriers to accessing health information could be viewed as insurmountable.

Mya described the integral role her sister-in-law had played during her pregnancy, driving her to appointments, translating during appointments and while she was labouring in hospital. With the support of her sister-in-law, she felt well informed throughout her pregnancy.

One Karen woman, SuSu, explained that, for her family and in her Karen culture, the management of health needs and seeking of health information was communal. Thus, when her grandmother needed health information, family members attended her doctor’s appointments with her. SuSu recalled feeling unwelcome andtension with the doctor when her grandmother was diagnosed with cancer. She explained:
*…when my, my grandma was diagnosed with cancer, she prefer a family member… to go with her. It’s more comfort for her… but then the, the doctor prefer like a professional translator, not the family member… So, like, it becomes like a problem…they prefer like a family member not to be there I think… they don’t want like the influence of the family member… that’s like difficult you know, especially for my grandmother… she’s vulnerable you know, and sometimes she couldn’t make the decision. And then there’s like, I feel like there’s a pressure, like in the room, like she doesn’t understand things properly.*(SuSu)

For some, the Church was also a crucial part of their community network, offering a source of health information through community information sessions, but also a sense of spiritual healing and safety.


*Okay, oh I got some information uh, from the training that uh, I attended at Church… how to stay healthy, of sort of exercise supposed to do and then what sort of food make us more healthy and then happy…*
(NilarI)

As women became more experienced and felt more proficient in accessing health information and health services, they sought opportunities to support other members of their communities, particularly new arrivals. Chaw explained that her experiences,

*…really motivate me to help the newly arrival who doesn’t know anything especially when they are pregnant. So I, I really want to try to help them…*.(ChawI)

### 3.3. Superordinate Theme: Seeking Health Information in the Context of Past Experiences

#### 3.3.1. Subordinate Theme: Health Information as a By-Product of Healthcare

Women were asked to reflect on experiences prior to resettlement in which they required or sought health information in Myanmar, Malaysia and while living in refugee camps. Often women responded that they did not have experiences to share or they described experiences in which healthcare and health information were not available to them. Chaw recalled,
*…I grew up in this very small village, there’s no hospital. And I never saw my parents go to the hospital either. That’s how I grew up and that is what my experience. And then I came to you know, uh, uh to Malaysia and then you know, continually you know, working… and then came to Australia. So I didn’t really have um, you know, the sort of mind knowledge about health.*(ChawI)

Khin and Tin (sisters) presented dental hygiene as an example of the absence of health information in their lives in Myanmar.


*Ah, it’s so different because um, since she was little like, doesn’t know dental care and every day, because they can’t afford to buy the brush even… So she first using the dental brush, uh, when she was fifteen.*
(TinB)

Experiences of seeking health information and healthcare were intertwined and typically reflected the hardships faced by women. Most often, however, women shared experiences of seeking healthcare without accompanying health information. Seeking healthcare was typically reserved for emergencies in which women were grateful to have survived. Nilar shared such an experience of becoming unwell during her pregnancy.


*…there was a Chin doctor uh, live in Malaysia, so I got uh, her number through friends and I just ring, I rang her and unless you come, I won’t probably make it… she did came… then you know… give me treatments…*
(NilarI)

Similar to experiences following resettlement, pregnancy and emergencies motivated women to seek health information or healthcare. For example, Thida sought health information during her pregnancy in Malaysia.


*…the family doctors in Malaysia will help her to make a bit of movement for the [unborn] child… and help her like, to do bit of exercise, what to do, what not to do…*
(ThidaB)

#### 3.3.2. Subordinate Theme: Health Professionals Provide Health Information

Many of the experiences that the women shared were of seeking healthcare or treatment rather than health information specifically. Asin the mannertheir experiences following resettlement in Australia, health professionals were presented as the primary source of health information for women in Myanmar, Malaysia and refugee camps. Dewi explained, *we just ask the doctor right way* (Dewi^B^). Aung, who expressed confidence using the internet to find information about health services on resettlement, offered more context.


*Now in Myanmar they have Internet, everything, they can search. When I wasthere, nothing you can do… Just go and see doctor, you explain to them and then they advise you what to do.*
(Aung)

Aung identified the Internet as a source of health information for her while she was living in Malaysia. This was a skill she brought with her and continued to use in Australia. However, she was the only participant who identified the Internet as a source of health information prior to resettlement.

Women living in refugee camps described readily accessible health services thatthey attended when they had healthcare or health information needs. Thida described this simply,


*…in Malaysia, like a U.N. operated clinic…it’s free of charge… she can go and ask anything she wants to know.*
(ThidaB)

#### 3.3.3. Subordinate Theme: Accessibility of Healthcare

Similar to their stories of accessing health information in Australia, women presented health professionals as their primary source of health information prior to resettlement. Barriers and facilitators to healthcare and thus, health information centred around the accessibility of health services and health professionals.

Nilar articulated the barriers she faced seeking health information and healthcare during her pregnancy in Malaysia. Similar to experiences on resettlement in Australia, family, friends and community assisted women to overcome barriers to healthcare and acted as sources of information themselves in refugee camps and Malaysia. Nilar recounted,


*…in Malaysia, we’re living really you know like, our living situation is very bad. If we go to the doctor, you know, we don’t speak their language and uh, the other problem is, it cost money…we go to ah, the Chin family, and she came and give me, give me the treatment for free.*
(NilarI)

Unlike experiences in Australia, the cost of seeing a health professional was a major barrier to health information and healthcare in Myanmar and for women living in Malaysia. Aye Aye described how cost could be prohibitive in Myanmar:


*So if um, someone’s not affordable, we just ask around and just get the treatment with the traditional or whatever you know, the community use. Um, if you’re affordable, and you can see the GP and then go to the hospital as required.*
(Aye AyeB)

In contrast, women who lived in refugee camps emphasised that free access to health services in refugee camps was a facilitating factor in the use of these services. One woman, Phyu^B^, shared her insights as a midwife in the refugee camps. She reported how health professionals identified and addressed women’s health needs, seeking women out how to offer them immunisations and health information. When checking on new mothers,

*…she also need to educate the mother about nutrition. And so, if they need help withthe immunisations as well, she needs to go and help*. (PhyuB)

#### 3.3.4. Subordinate Theme: Expectations on Resettlement

Women’s experiences of healthcare, health information or the absence of these experiences influenced women’s expectations on resettlement. Despite the role of the GP as women’s primary source of health information on resettlement in Australia, Phyu explained that GPs were a new concept:


*…in Myanmar we don’t have like a family GP. We can’t, we don’t even use GP at all in our country. So she was not aware of even the term, GP, when she arrived.*
(PhyuB)

Caseworkers and family members introduced women to the concept of a GP. Because of the support of caseworkers and relatives, women emphasised that seeking health information in Australia and learning how to navigate the Australian health system was not a struggle. However, navigating the health system emerged as a barrier to health information on resettlement.

Medications and health interventions could be unfamiliar to women. Win, a young Karen woman who arrived in Australia as a child, described the volume of health information on arrival as overwhelming. Because they received so much health information, she reflected, *…it’s just forgetful when you keeping track of a lot of information in one* (Win). She reported that they did not understand much of the information offered to them and hence, did not take the medications and supplements offered to them during their initial screening.

When she became a mother herself, SuSu described being caught between the information and advice she received from her family members and health professionals. Balancing health information from these two sources could cause be stressful. SuSureflected on the information offered by family members as contextual, but no longer appropriate on resettlement in Australia. She recalled an instance when her child was sick:


*…they would lose a lot of their kids due to, you know sickness and, just not having the right or enough… medication or, have the right diagnose [sic]. Um, so like for that reason, it’s very panicky for them, you know every time their kids or grandkids are sick. They would just… they want to give them all this medication that they use to but like here, you can’t… even get like those infection, like medication.*
(SuSu)

## 4. Discussion

This study offers novel and valuable insights into the experiences of women of refugee background from Myanmar seeking health information. Interpretative phenomenological analysis revealed that women were motivated to seek health information forpregnancy and illness. Communication and navigating the health system emerged as barriers to accessing health information, reflecting women’s preference to access health information from health professionals. However, the community emerged as a powerful resource upon which women drew to manage their health information needs. Finally, women’s experiences of seeking health information were understood in the context of pre-migration experiences, which influenced their expectations and behaviours on resettlement. These valuable insights can be used to enhance women’s access to health information and their health literacy. This discussion will explore how the women’s experiences relate to their health literacy and potential opportunities co-designed and co-facilitated by peers to improve women’s access to and understanding of health information, or their health literacy, in ways that are informed by the women’s cultural and trauma-informed needs, and prior health experiences: Internet-based resources, interactions with health professionals, antenatal care and community-based education.

### 4.1. Health Literacy

While this study sought insight into the women’s experiences of seeking health information, these experiences were described in the context of how this information was obtained, understood and used. Women reflected critically on what they had learnt from these experiences, and how these experiences reflected their broader hardships without prompt. These experiences can be considered against the definition of health literacy provided earlier, “the literacy and numeracy skills that enable individuals to obtain, understand, appraise, and use information to make decisions and take actions that will have an impact on health status” [18] (p. 161). A model of how the women’s experiences related to this definition of health literacy is shown in Figure 1.

This model shows that seeking heath information began with a motivation. As women developed confidence and knowledge from experiences seeking health information on resettlement, they sought out opportunities to support family and community members, particularly new arrivals; their motivation to obtain health information moved away from meeting immediate health needs to supporting the needs of community members. Women navigated barriers not only to seeking health information, but also to processing, understanding and communicating about it. Health literacy skills can be developed and enhanced at the individual and community levels [18]. Insight into the women’s motivations to seek health information and their experiences of doing so can be used to tailor interventions to improve their health literacy skills. Research into those health literacy interventions, including those which may improve health communication, has been recommended [18].

### 4.2. Enhancing Community Health Literacy

Community was revealed as an important salutogenic resource in this study. Peers, family members and community leaders supported women to access and understand health information by identifying and introducing women to health services, sharing health information, translating health information and assisting with practical aspects, such as transport. As women gained health knowledge and developed health literacy skills, they were eager to share these and support other members of their community. Collectivism has previously been identified as facilitating the health service access of women of refugee background from Myanmar [13,37]. This strong community network and collective approach to managing health needs is a resource thatmay be enhanced to improve the health knowledge and health literacy of the community as a whole.

The successful implementation of community health education programs designed specifically for women of refugee background has been reported in the literature. Svensson et al. [19] reported on a program of sexual and reproductive health education delivered by peer health educators in Sweden to women of refugee background from Afghanistan, Iraq, Somalia and Iran. The peer health educators completed training prior to delivering the education program. They were able to deliver culturally sensitive health information in the women’s preferred language with an understanding of the women’s pre-migration experiences. The women who participated in the programs felt better informed, more confident and motivated to learn more following their participation in the education program. Furthermore, they shared the information they learnt with their peers outside of the program. Similarly, Frost [38] reported on a community education program trialled with women of refugee background from Myanmar resettled in the United States. This program was also co-facilitated by health educators of refugee background. The purpose of this program was to offer more general health information, including information on how to access and navigate health services. Both of these programs successfully engaged participants, enhanced community health knowledge and communicated health information in a culturally safe way in the women’s preferred languages [19,38]. Such community education programs have the potential to enhance the pre-existing resourcefulness and community networks demonstrated by women of refugee background from Myanmar in this study.

### 4.3. Optimising Interactions with Health Professionals

This study identified health professionals as the preferred source of health information for women of refugee background from Myanmar. GPs were specifically frequently cited as the women’s primary point of contact with health services. GPs have previously been recognised as key figures in the health and resettlement of people of refugee backgrounds in Australia and the United States [39,40]. Barriers associated with accessing health services, that is, difficulty navigating the health system and communicating with health professionals, emerged as major barriers to accessing health information for women of refugee background from Myanmar.

Effective communication is essential to obtaining and understanding health information [39]. Women of refugee background from Myanmar shared difficulties faced when they could not express themselves in English or an interpreter was not arranged in their preferred language. Although overcoming language barriers and engaging appropriate interpreters can be significant challenges, effective cross-cultural communication is more complex than the provision of interpreters [40,41]. A Swedish study of communication between physicians and patients from Iran revealed body language was a significant aspect of insufficient communication [41]. Similarly, the women of refugee background from Myanmar in this study were alert to health professionals’ body language and any indications that they may not be welcome.

As survivors of trauma, people of refugee background may perceive discrimination and a power differential of whichthe health professional is not cognisant [39]. Thus, cultural sensitivity and safety and an understanding of trauma is essential for effective communication. Health professionals need to be supported with effective training to communicate with and meet the health information needs of people of refugee background. Training has been recommended to support health professionals to improve their cross-cultural communication skills [19]. Im and Swan [42] reported on the trial of culturally responsive trauma-informed training workshops for mental health professionals in the United States. Leaders of refugee background communities co-designed and co-facilitated these workshops, facilitating intercultural learning and relationships between community leaders. Participation in culturally responsive trauma-informed training may support health professionals to develop an improved understanding of the health needs of refugee background women from Myanmar, improved cross-cultural communication skills and improved relationships with community leaders.

### 4.4. Engaging Women during Pregnancy

Prior to and following settlement, pregnancy emerged as a common motivation for seeking health information and accessing health services for women of refugee background from Myanmar. Pregnancy is typically a woman’s first interaction with health services on resettlement in a new country [43]. The women’s experiences of antenatal complications revealed a clear need for education on routine antenatal care, the identification of such complications and how to access emergency health services if needed. These experiences echo prior literature reporting higher rates of complications of pregnancy and birth amongst women of refugee background [44].

Pregnancy presents an ideal opportunity to engage women (and their family members) with health services, offer health information and to address social determinants of health, such as health literacy [45]. This study showed that women of refugee background from Myanmar have unique needs of antenatal care thatmay be informed by culture, trauma and prior experiences of pregnancy and healthcare. A community-based model of group pregnancy care co-designed with Karen women from Myanmar delivered by a multidisciplinary team, including a caseload midwife and a maternal and child health nurse, has been shown to successfully engage Karen women [44]. This model facilitated improved access to pregnancy information and health services through the involvement of a bicultural worker, communication in the women’s preferred language, shared learning with peers in group education sessions and continuity of health professionals. Riggs et al. [44] reported that women felt informed and more confident as a result of this group pregnancy care model. Such models of care may be ideal for engaging women during pregnancy and meeting their specific health needs.

### 4.5. Internet-Based Resources

The Internet emerged as a source of health information on resettlement for the women in this study. Similarly, Kaneoka and Spence [46] identified the Internet as a source of sexual and reproductive health information for women of refugee and asylum seeker backgrounds from the Middle East. However, these women had pre-migration experiences of using the Internet for health information. Only one woman in this study reported a pre-migration experience of using the Internet to search for health information. Other women in this study were motivated and demonstrated an ability to learn how to use the Internet to meet their health information needs. The successful use of Internet-based resources requires e-health literacy skills, that is, health literacy skills in the context of digital technology use [47]. Further research into the e-health literacy of women of refugee background from Myanmar and supporting them to develop their e-health literacy skills is warranted.

Internet- and mobile phone-based applications (apps) have been developed in response to the health needs of people of refugee backgrounds [48,49]. Apps have the advantage of bypassing barriers related to accessing health services, such as making appointments or cost [45]. However, the successful use of Internet-based applications can be inhibited by the level of education, language proficiency and literacy [49]. Bartlett et al. [48] reported the development of an app to facilitate the access of English- and Arabic-speaking women of refugee or migrant backgrounds to sexual and reproductive health information. The app was co-designed with women of refugee and migrant backgrounds to ensure it was culturally safe and met their information needs. The development of apps to facilitate the access of women of refugee background from Myanmar to health information warrants further exploration and requires rigorous evaluation.

### 4.6. Strengths and Limitations

This study offered novel insight into the experiences specifically of women of refugee background from Myanmar seeking health information on resettlement. Hence, these findings may be of limited generalisability to other populations, but they can be related to other refugee women groups in immigrant nations. Considerable effort was made to ensure that the women felt comfortable and safe to share their experiences, including the presence of the bicultural researcher, the woman’s choice of location and the interpreter, bicultural researcher and first author all being of female gender. A strength of this study is that women were interviewed in their preferred language, enabling women to participate who may not otherwise have the opportunity to participate in such research. To do so, an interpreter was engaged for some women, and a bicultural researcher co-facilitated interviews and interpreted for other women. The same interpreter was engaged for all of the interviews with the Chin women for continuity. Language and communication are fundamental to data collection and analysis in cross-cultural qualitative research [50]. Thus, the roles of the bicultural researcher and interpreter, and the treatment of data collected in Burmese and Chin have been reported with transparency to enhance the rigour of the research. Bicultural researchers and interpreters contribute their own cultural knowledge to interpretation and data collection. Thus, the first author reflected on the interviews with the bicultural researcher and interpreter to facilitate reflexivity. This enhances rigour in IPA [31].

## 5. Conclusions

This study offered key insights into women’s health information experiences. These experiences revealed opportunities to enhance women’s access to health information, engagement with health services and support the development of their health literacy and health knowledge. Women’s experiences prior to resettlement informed how they sought health information on resettlement. Interventions co-designed and co-facilitated by peers, specifically trauma-informed culturally responsive training, group pregnancy care, apps and community education programs, were presented to ensure that women are supported to access health information in a way that is safe and respectful of their cultural, trauma-informed needs and prior health experiences. Pregnancy emerged as an area that required improved knowledge of and engagement with health services, while interactions with health professionals were ideal opportunities to engage women with improved communication. Women’s newfound Internet skills also offered an opportunity to improve their access to health information through co-designed health apps. Finally, women drew upon strong community networks to navigate around barriers to health information. Community education programs designed to improve aspects of women’s health knowledge, including their navigation of services, may be an effective approach to improve the health knowledge and health literacy of this population. Further research into the co-design and evaluation of such interventions with women of refugee background from Myanmar is recommended.

## Figures and Tables

**Figure 1 ijerph-19-03289-f001:**
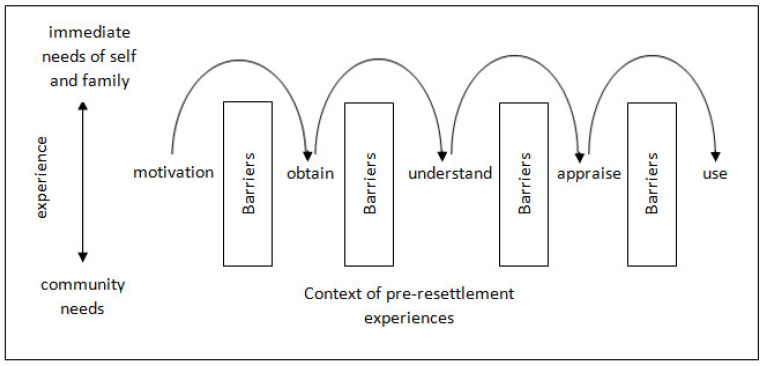
A model of the health information experiences of women of refugee background from Myanmar.

**Table 1 ijerph-19-03289-t001:** Participant demographics.

Participant (Pseudonym)	Ethnicity	Language(s) Spoken	Age (Years)	Years in Australia	Occupation	Level of Education Completed
Aye Aye	Kachin	Kachin, Burmese	39	8	Unemployed	TAFE
Dewi	Kachin	Kachin	32	5	Applying for a job	High School
Thida	Mon	Burmese	36	8.5	Not disclosed	Not disclosed
Aung	Kachin	Kachin, English	41	3	Study	TAFE
Nandar	Mon	Mon, English, Burmese	37	9	Paid work	TAFE
SuSu	Karen	Karen, English	26	15	Mother/student	University
Khin	Chin	Burmese, Chin	61	6	Unpaid carer for parents	Primary School
Tin	Chin	Burmese, Chin	55	6	Stay at home mother	Primary School
Win	Karen	English, Karen	25	14	Care Aide	TAFE
Nilar	Chin	Chin	49	12	N/A	TAFE
Chaw	Chin	Chin	32	13	Paid (Family Day Care)	High School
Thet	Chin	Chin	50	10	Housewife	Not disclosed
Mya	Chin	Chin	30	4	Housewife	Primary School
Phyu	Chin	Burmese	42	1 year 3 months	Unemployed	High School

**Table 2 ijerph-19-03289-t002:** Themes.

Superordinate Theme	Subordinate Theme
Seeking health information	Motivation
	Sources
Facilitators and barriers	Communication
	Navigating the system
	Community
Seeking health information in the context of past experiences	Health information as a by-product of healthcare
	Health professionals provide health information
	Accessibility of healthcare
	Expectations on resettlement

## Data Availability

The data presented in this study are available on request from the corresponding author.

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
