# Peer review of "Seeking Health Information: A Qualitative Study of the Experiences of Women of Refugee Background from Myanmar in Perth, Western Australia"

_ijerph, 2022, doi:10.3390/ijerph19063289_

Round 1

Reviewer 1 Report

The study titled "Seeking health information: A qualitative study of the experiences of women of refugee background from Myanmar in 3 Perth, Western Australia" analyses the challenges faced by Myanmar refugee women in seeking healthcare.

The study is an attempt to add to our growing understanding healthcare accessibility related problems encountered  by refugees.

Comment

This study needs a major revision and reformatting.

The abstract needs to be more clear about the superordinates and subordinates.

The differences between experiences and access should be explained better.

The author contribution must be acknowledged in a separate section instead of methods section.

The multiple referencing of an article in a section must be rectified.

Reviewer 2 Report

The article written by Griffin on women of refugee background has touched an important topic.

Although, it has raised important issues, there is a need of several refinements before it can get accepted.

  1. The abstract lacks focus, there is no result included, it looks like they were discussing about methods and ended up with recommendations.
  2. Since there are number of barriers in the process of interviews conducted with the participants, there is a need for detailed description of steps taken for each case to prevent bias. The write up given currently is not easy to understand and correlation is difficult. I proposed a table for all participants with descriptions.
  3. I think, they should have done some inferential analysis to understand the impact of some independent factors on the health seeking information.
  4. Authors need to specify about the tool they used for conducting interviews, what steps were taken to keep consistency on the inquiring items? There is need to present it in the form of a table or schematic chart.
  5. What do you mean by group pregnancy care model?
  6. Authors need to add a description about the kind the facilities available in the refugee to get to know in what context this study was done.
  7. 5 section speaks about internet-based resources. that means they had internet facilities.

Overall, there is a need to proof reading of the article to make it more understandable.

Reviewer 3 Report

Manuscript Number: IJERPH-1598709

Title: Seeking health information: A qualitative study of the experiences of women of refugee background from Myanmar in Perth, Western Australia

Thank you for your interesting report. This report presented “Seeking health information: A qualitative study of the experiences of women of refugee background from Myanmar in Perth, Western Australia.” I have some suggestions for revision.

<Minor revision>

  1. “Women were motivated to seek health information by need. Such needs could include pregnancy, feeling unwell including chronic and complex health needs, and the needs of children or other family members (page 7 line 269-271).” I wonder if there are any common diseases they come across. Also, I wonder if the pattern of finding health information varies depending on the disease, such as acute infectious disease and diseases that require lifestyle modification such as diabetes or high blood pressure.

Author Response

Thank you for your comment. Please see the attachment. 

Round 2

Reviewer 1 Report

I had no serious reservations about the manuscript previously and they have addressed my concerns, so it would be ok if the manuscript is accepted for publication.

Reviewer 2 Report

Authors have done required changes in the manuscript.

Reviewer 3 Report

Thank you for your effort.